# Hyperbolic photonic topological insulators

Lei Huang [1,2,5], Lu He [1,2,5], Weixuan Zhang [1,2,5] ✉, Huizhen Zhang[1,2], Dongning Liu [3], Xue Feng [3], Fang Liu[3], Kaiyu Cui [3], Yidong Huang[3,4], Wei Zhang [3,4] ✉ & Xiangdong Zhang [1,2] ✉

Topological photonics provides a new degree of freedom to robustly control electromagnetic fields. To date, most of established topological states in photonics have been employed in Euclidean space. Motivated by unique properties of hyperbolic lattices, which are regular tessellations in non-Euclidean space with a constant negative curvature, the boundary-dominated hyperbolic topological states have been proposed. However, limited by highly crowded boundary resonators and complicated site couplings, the hyperbolic topological insulator has only been experimentally constructed in electric circuits. How to achieve hyperbolic photonic topological insulators is still an open question. Here, we report the experimental realization of hyperbolic photonic topological insulators using coupled ring resonators on silicon chips. Boundary-dominated one-way edge states with pseudospin-dependent propagation directions have been observed. Furthermore, the robustness of edge states in hyperbolic photonic topological insulators is also verified. Our findings have potential applications in the field of designing high-efficient topological photonic devices with enhanced boundary responses.

Engineering photonic nano/micro-structures with non-trivial topologies has been receiving a lot of attention in recent years[1,2], and is believed to be a key method for the realization of disorder- and defect-immune photonic devices. Analogy to condensed matter physical systems, a large number of fascinating topological states have been successively proposed in various photonic systems with different characteristics. For example, topological photonic insulators and semimetals have been experimentally realized[3–22]. Exotic phenomena in topological photonic quasicrystals and fractal insulators have been fulfilled using coupled optical waveguide arrays[23,24]. In addition, the non-Hermitian and non-linear topological states are also widely exploded in photonics[25–27], giving important platforms for the construction of functional devices with robust performances, such as topological lasers[28–34], topological frequency combs[35] and topological sources of quantum light[36]. However, based on the bulk-boundary correspondence, the $n$th-order topological phases are always featured by boundary states with $n$-dimensional lower than the bulk that hosts

them. In this case, the volume of topological channels is much smaller than trivial bulk domains, limiting the utilization efficiency of topological structures in photonics. Hence, if we can construct topologically protected edge states in photonic systems with a much larger number of boundary sites than that of bulk sites, the efficiency of topological photonic devices is expected to be significantly improved.

On the other hand, the non-Euclidean geometry widely exists in natural and artificial systems[37], and plays important roles in different fields. The recent ground-breaking implementations of two-dimensional hyperbolic lattices in circuit quantum electrodynamics[38] and topolectrical circuits[39] have stimulated numerous advances in hyperbolic physics, including the hyperbolic band theory[40,41], the crystallography of hyperbolic lattices[42], quantum field theories in continuous negatively curved spaces[43], the Breitenlohner-Freedman bound on hyperbolic tiling[44], highly degenerated hyperbolic flatbands[45,46], many-body hyperbolic models[47] and so on[48–52]. Beyond those exotic physical phenomena, there are many investigations on

[1]Key Laboratory of advanced optoelectronic quantum architecture and measurements of Ministry of Education, Beijing Institute of Technology, 100081 Beijing, China. [2]Beijing Key Laboratory of Nanophotonics & Ultrafine Optoelectronic Systems, School of Physics, Beijing Institute of Technology, 100081 Beijing, China. [3]Frontier Science Center for Quantum Information, Beijing National Research Center for Information Science and Technology (BNRist), Electronic Engineering Department, Tsinghua University, Beijing 100084, China. [4]Beijing Academy of Quantum Information Sciences, Beijing 100193, China. [5]These authors contributed equally: Lei Huang, Lu He, Weixuan Zhang. ✉e-mail: zhangwx@bit.edu.cn; zwei@tsinghua.edu.cn; zhangxd@bit.edu.cn

the construction of hyperbolic topological states[53–61]. Hyperbolic topological band insulators with non-trivial first/second Chern numbers and hyperbolic graphene have been theoretically created and experimentally realized[60,61]. In addition, the robust one-way propagation of boundary-dominated hyperbolic Chern edge states was also fulfilled by electric circuit networks[56]. It is important to note that boundary sites always occupy a finite portion of total sites regardless of the size for the hyperbolic topological lattice. Hence, if we can construct such boundary-dominated hyperbolic topological states in photonic systems, the topological photonic devices with enhanced edge responses are expected to be achieved. However, limited by the requirement of crowded boundary resonators and complicated site couplings, the experimental realization of hyperbolic photonic topological insulators is still lacking.

In this work, we give the experimental demonstration on the realization of hyperbolic photonic topological insulators by coupled optical ring resonators. We design and fabricate face-centered and vertex-centered hyperbolic topological insulators with non-trivial real-space Chern numbers using silicon photonics. By measuring transmission spectra and steady-field distributions of hyperbolic photonic topological insulators, the boundary-dominated one-way edge states with pseudospin-dependent propagation directions have been observed. Furthermore, the robust edge propagations in hyperbolic photonic topological insulators with defects are also verified. Our work may have potential applications in designing high-efficient topological photonic devices with significantly reduced bulk domains.

## Results

### The theory of hyperbolic photonic topological insulators

We consider a $\{6, 4\}$ hyperbolic lattice in the Poincaré disk, where the center of a hexagon locates at the origin, as shown in Fig. 1a. We call this lattice model as the face-centered hyperbolic lattice. The Schläfli notation $\{6, 4\}$ represents the tessellation of a 2D hyperbolic plane by 6-sided regular polygons with the coordination number being 4. For clarity, the lattice sites in the first-, second-, and third-layer are represented by red, blue, and pink dots, respectively. The coupling patterns inside all hexagons can be divided into two categories. Nearly a half number of hexagons (marked by triangles) possess the nearest-neighbor (NN) hopping of $Je^{i\varphi/3}$, the next-nearest-neighbor (NNN) hopping of $Je^{i\varphi/6}$ and the next-next-nearest-neighbor (NNNN) hoppings of $J$, as shown in Fig. 1b. The remained half of hexagons only contain the NN coupling of $Je^{i\varphi/3}$, as shown in Fig. 1c. In this case, the hyperbolic lattice model can be effectively described by a tight-binding Hamiltonian as:

$$\hat{H} = \sum_i \omega_0 a_i^\dagger a_i + \sum_{<i,j>} Je^{i\varphi/3} a_i^\dagger a_j + \sum_{<<i,j>>_{half}} Je^{i\varphi/6} a_i^\dagger a_j + \sum_{<<<i,j>>>_{half}} Ja_i^\dagger a_j + h.c..$$

(1)

with $a_i^\dagger(a_i)$ being the creation (annihilation) operator at site $i$. $\omega_O$ is the on-site potential of each lattice site. The bracket $<\cdots>$ indicates the summation being restricted within all NN sites of the $i$-th site. Other two brackets $<<\cdots>>_{half}$ and $<<<\cdots>>>_{half}$ correspond to summations being restricted within a half number of NNN and NNNN sites of the $i$-th site.

It is noted that the complex-valued NNN couplings can create the staggered flux into a half number of hexagons, that can break the time-reversal symmetry of the system and introduce non-trivial topologies. The calculated eigenspectra ($\varepsilon$) of the three-layer hyperbolic model with $\varphi$ equaling to $\pi$ and $\pi/2$ are presented in two left charts of Fig. 1d, e. Other parameters are set as $J = 1$ and $\omega_0 = 0$. The color map represents the localization strength of eigenstates on lattice sites at the third layer, that is quantized by $V(\varepsilon_n) = \sum_{i \in L=3} |\boldsymbol{\phi}_i(\varepsilon_n)|^2 / \sum_{i \in L=[1,3]} |\boldsymbol{\phi}_i(\varepsilon_n)|^2$ with $\boldsymbol{\phi}_i(\varepsilon_n)$ being the

eigenmode at $\varepsilon = \varepsilon_n$. It is shown that there are a large number of eigenstates exhibiting boundary-localized spatial profiles. To further determine the topological properties of these edge states, we calculate the corresponding real-space Chern numbers shown in two right charts of Fig. 1d, e. It is shown that the non-zero platform of the real-space Chern number appears around the eigenenergy of $\varepsilon = 0$ ($\varepsilon = -1$) with $\varphi = \pi$ ($\varphi = \pi/2$), indicating the existence of Chern-class topological edge states in our designed hyperbolic lattices. It is worth noting that the calculated real-space Chern number is much closer to one than that of previously proposed $\{6,4\}$ hyperbolic Haldane model with the same number of lattice sites[49], showing a good superiority of our designed hyperbolic topological lattice model. In Supplementary Figs. S1 and S2, we give detailed numerical results on spatial profiles and robust one-way propagations of hyperbolic topological edge states. These results clearly show that non-trivial topological edge states exist in our designed face-centered hyperbolic topological lattice with suitably engineered staggered flux.

In fact, we can design the photonic nano-/micro-structures to realize the control of electromagnetic fields using boundary-dominated hyperbolic topological states. For this purpose, we construct the $\{6,4\}$ hyperbolic topological lattice by evanescently coupled optical resonators, where the schematic diagram of designed optical structure with two layers is shown in Fig. 1f. In this structure, optical ring resonators can be divided into the site rings (red and blue blocks in the first and second layers) and linking rings (the black blocks) according to their functionalities. Specifically, different site rings have exactly the same geometric parameters, ensuring the same resonant frequency and free spectral range (FSR) of all site rings. The suitably designed linking rings are used to couple different site rings to implement required site couplings. Owing to the aperiodicity of hyperbolic lattices, the size and coupling pattern of linking rings should be suitably designed. Here, the small-size linking ring is used to couple two boundary-site rings to simulate NN hoppings. In addition, six site rings are coupled by a single large-size linking ring to realize required NN, NNN, and NNNN couplings. In this system, we can tune the effective coupling strength $J$ between two site rings by tuning the separation distance between site rings and the link ring. In addition, the coupling phases between NN, NNN, and NNNN site rings can be adjusted by engineering the phase accumulation of the wave propagating from one site ring to the other through the link ring (See Supplementary Figs. S3–S5 for details). Geometric parameters of large coupling rings are illustrated in Fig. 1f. Figure 1g presents detailed parameters of radius and widths for site rings and small-size linking rings, as well as the distance between linking rings and site rings. Through the appropriate setting of spatial positions and coupling patterns of site rings and linking rings, the eigenequation of our designed evanescently coupled ring-resonator array is identical with that of the topological hyperbolic lattice model. In particular, the effective site energy of each resonator is $\omega_0 = 193.45 THz$. And, the amplitude and phase of effective site couplings equal to $J = 0.05 THz$ and $\varphi = \pi$. In this case, our designed face-centered hyperbolic photonic topological insulator (FHPTI) should possess non-trivial edge states around the central frequency in each FSR of site rings. In addition, it is worth noting that all eigenmodes of a single ring resonator can be categorized into two decoupled subspaces, namely clockwise modes and counterclockwise modes. The coupling strengths between clockwise and counterclockwise modes are extremely weak in our system. Consequently, we can consider clockwise and counterclockwise modes as a pair of decoupled pseudo-spins. In the following, we call these two pseudo-spins as the clockwise pseudo-spin (CPS) and the anticlockwise pseudo-spin (APS). Two pseudo-spins can be suitably excited from two different ports (shown in Fig. 1f). Importantly, it should be emphasized that the effective NN, NNN, and NNN coupling phases in counterclockwise and clockwise subspaces are conjugate to

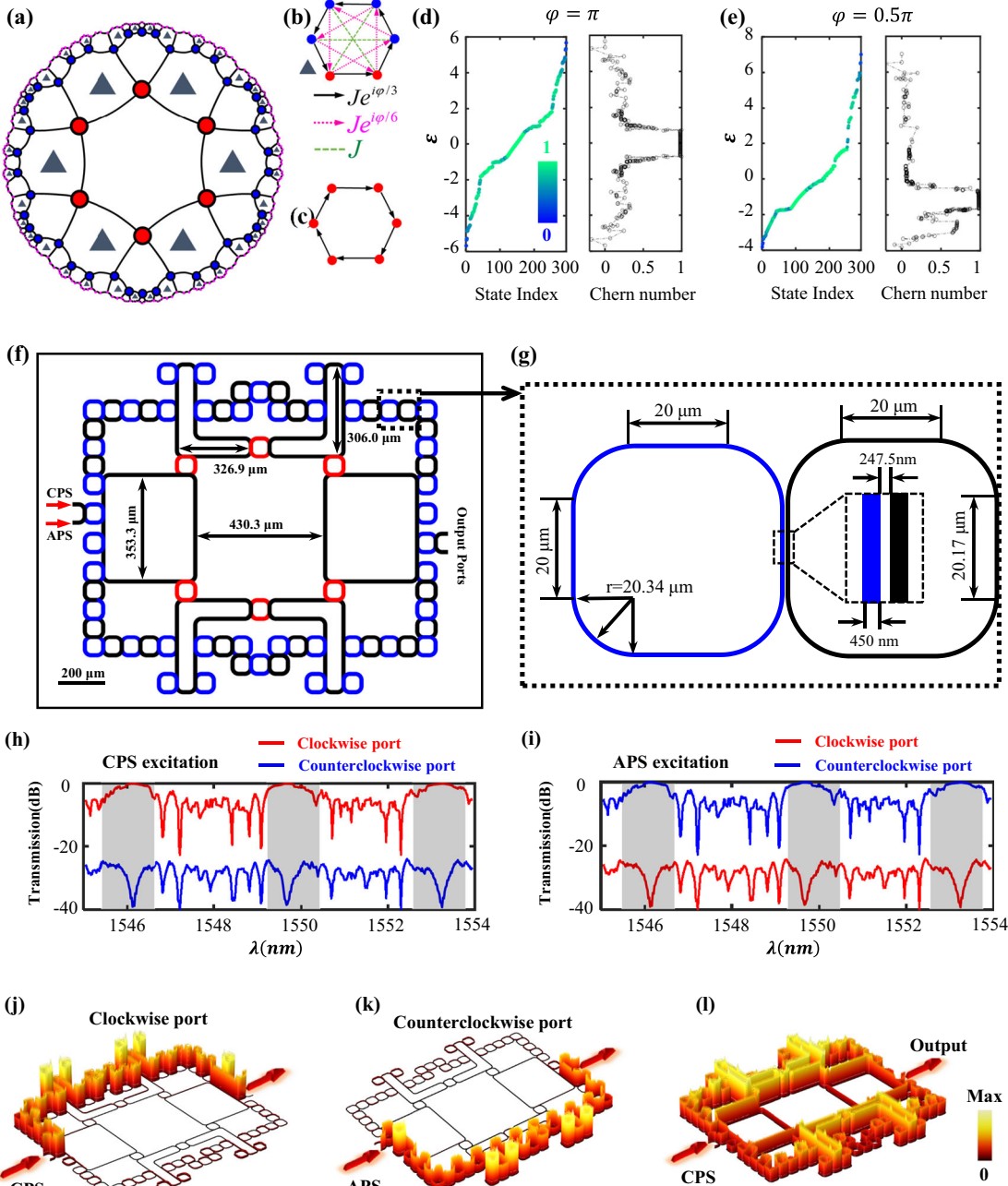

**Fig. 1 | Theoretical results of hyperbolic photonic topological insulator in the face-centered {6, 4} hyperbolic lattice. a** The illustration of face-centered {6, 4} hyperbolic lattice in the Poincaré disk. Lattice sites in the first, second, and third layers are represented by red, blue, and pink dots. Here, a half number of hexagons contain the staggered flux, as marked by triangles. **b** The coupling pattern in the hexagon possessing *NN, NNN,* and *NNNN* hoppings. **c** The coupling pattern in the hexagon only possessing *NN* hoppings. **d, e** The eigenspectra and real-space Chern numbers of the three-layer face-centered hyperbolic model with φ equaling to π and π/2, respectively. **f** The schematic diagram of the designed hyperbolic photonic topological lattice with two layers. **g** The illustration of detailed geometric parameters, including radius and widths of site and small-linking rings and the distance between linking rings and site rings. **h, i** Simulated transmission spectra by exciting the CPS and APS, respectively. Red and blue lines correspond to results of output ports at clockwise and counterclockwise positions with respect to the input port, respectively. **j, k** Simulation results of steady-state distributions of electric fields by exciting the CPS and APS with the wavelength being 1549.7 nm. **l** The steady-state distribution of electric fields with the excitation wavelength being 1550.3 nm.

each other. Therefore, the calculated Chern numbers associated with counterclockwise and clockwise subspaces exhibit opposite signs.

To further demonstrate exotic topological effects, we perform the full-wave simulation of wave propagation in FHPTI using finite element methods. It is important to note that the two-dimensional (2D) simulation results can effectively predict the performance of real 3D optical structures by setting the appropriate optical parameters in the system. Here, the effective refractive index of the optical waveguide

(environment) is set as 2.832 (1.2) to simulate the 3D silicon waveguides embedding into the background of silica (See Supplementary Fig. S6 for details). Figure 1h, i displays the simulated transmission spectra by exciting the CPS and APS, respectively. Red and blue lines correspond to results from output ports at clockwise and counterclockwise positions with respect to the input port, respectively. Around the central frequency of each FSR (highlight by dark regions), there is a large transmission platform from the clockwise

(counterclockwise) output port under the CPS (APS) excitation, showing the existence of a pseudospin-dependent topological edge state. Away from the central frequency, a large number of transmission peaks appear and transmissions of the clockwise (counterclockwise) output port under the excitation of CPS (APS) are significantly decreased, indicating the excitation trivial bulk eigenmodes. Figure 1j, k presents the steady-state distributions of electric fields by exciting the CPS and APS at 1549.7 nm (equaling to the central wavelength of $2\pi c_0/\omega_0$). It is shown that the one-way transport of input signals along the edge with pseudospin-dependent propagation directions appears, showing key behaviors of topological edge states. For comparison, we also calculate the steady-state distribution of electric fields by exciting the bulk state at 1550.3 nm, as shown in Fig. 1l. It is shown that the input electric fields can permeate into the bulk, and the bidirectional edge propagation also appears, meaning the excitation of trivial bulk and edge states. These simulation results demonstrate the correctness on the implementation of hyperbolic topological insulators by coupled optical-ring resonators.

Except for the above design of FHPTIs, in the following, we show that the photonic topological insulators can also be designed in hyperbolic lattices with a vertex locating at the center of the Poincaré disk. We call such lattice model as the vertex-centered {6, 4} hyperbolic lattice. Figure 2a displays the vertex-centered hyperbolic topological lattice. Details of the corresponding topological properties are provided in Supplementary Figs. S7 and S8.

Then, we use evanescently coupled optical ring resonators to construct the vertex-centered hyperbolic photonic topological insulator (VHPTI) with three layers, as shown in Fig. 2d. We note that the number of site rings (equaling to 113) for the three-layer VHPTI is much larger than that of the two-layer face-centered counterpart (equaling to 48). In addition, the $C_6$ symmetry of face-centered structure is reduced to the $C_2$ symmetry for VHPTIs. Hence, a greater number of large-size linking rings are required to implement the VHPTI. It is worth noting that all parameters of site rings and small-size linking rings are identical with that used in the FHPTI. The sizes of large linking rings are displayed in Fig. 2d. In this case, the effective parameters of the VHPTI are still equaling to $\omega_0 = 193.45 THz$, $J = 0.05 THz$ and $\varphi = \pi$.

The calculated transmission spectra of the VHPTI under CPS and APS excitations are shown in Fig. 2e, f. Similar to the face-centered counterpart, there is a large transmission platform of the clockwise (counterclockwise) output port with the CPS (APS) being excited. And, the transmission of clockwise (counterclockwise) output port is significantly decreased in the frequency range away from the central frequency under the CPS (APS) excitation, corresponding to the excitation of trivial bulk/edge states. The calculated steady-state distributions of electric fields are shown in Fig. 2g, h under excitations of CPS and APS at 1549.7nm. It is found that the electric fields can unidirectionally propagate along the boundary of the structure, and the propagation direction depends on the input pseudospin. In addition, during the propagation, the electric field is confined to the boundary and does not penetrate into the bulk. These simulation results clearly prove the existence of unidirectional boundary states in the VHPTI. In contrast, there are significant electric fields in the bulk region when the trivial bulk state is excited at 1550.3nm, as shown in Fig. 2i. Full wave simulations clearly prove the correctness of our design. It is worthwhile to note that ratios between the number of boundary optical

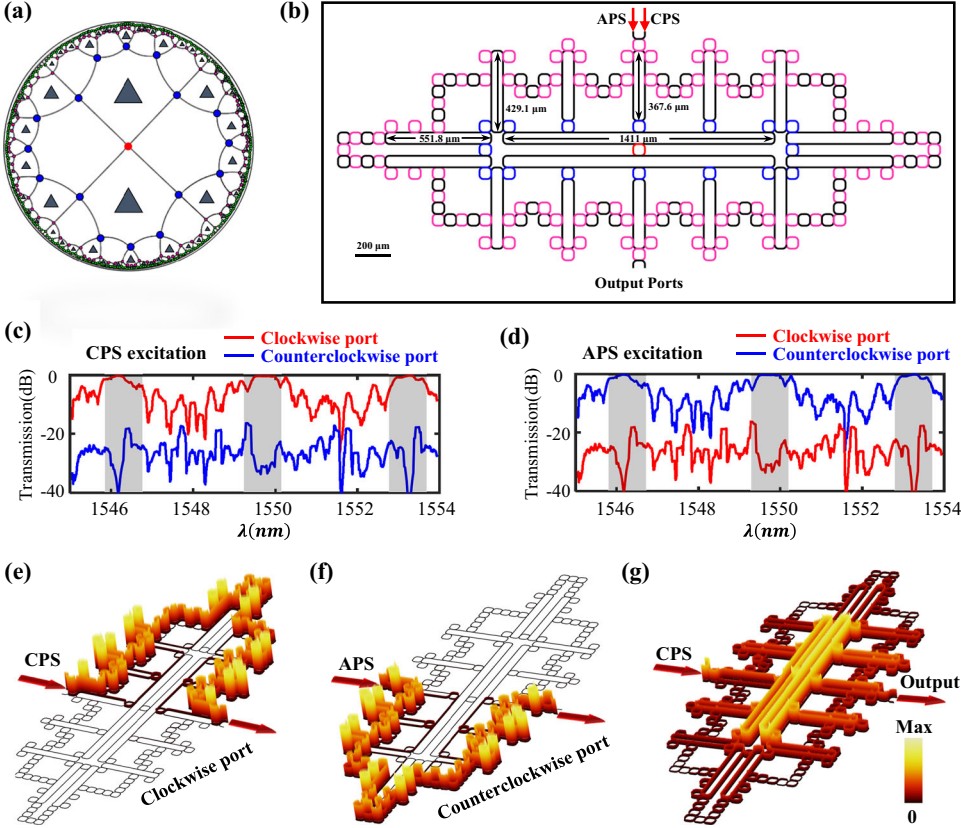

**Fig. 2 | Theoretical results of hyperbolic photonic topological insulator in the vertex-centered {6, 4} hyperbolic lattice. a** The illustration of vertex-centered {6, 4} hyperbolic lattice in the Poincaré disk. Lattice sites in first, second and third layers are represented by red, blue and pink dots. Here, a half number of hexagons contain the staggered flux, as marked by triangles. **b** The schematic diagram of the designed vertex-centered hyperbolic photonic topological lattice with three layers. Detailed geometric parameters are presented. **c, d** Simulated transmission spectra by exciting the CPS and APS, respectively. **e, f** Simulation results of steady-state distributions of electric fields by exciting the CPS and APS with the wavelength being 1549.7 nm. **g** The steady-state distribution of electric fields by exciting the trivial bulk states at 1550.3 nm.

resonators and that of optical resonators in the bulk are about 0.875 and 0.8496 for the two-layer FHPTI and three-layer VHPTI. They are much larger than the Euclidean counterparts (0.4375 and 0.3306) with same numbers of site rings. Hence, such an enhanced boundary response can improve efficiencies of some next-generation topological photonic devices.

## Experimental observation of hyperbolic photonic topological insulators by evanescently coupled optical resonators

In this part, we experimentally demonstrate the realization of hyperbolic photonic topological insulators. The above designed optical structures are fabricated on a 220 nm-thick Si layer coated on the $SiO_2$ substrate using electron-beam lithography followed by plasma etching (See Method for details). To maintain the up-down symmetry of the structure, the sample is coated with a layer of $SiO_2$. The microscopy image of the fabricated FHPTI is shown in Fig. 3a. The enlarged view is displayed in Fig. 3b. It is shown that there is a good consistence between the fabricated structure and the theoretical design. In experiments, to demonstrate topological properties of the sample, we measure the frequency-dependent transmission spectra. Here, the CPS and APS can be selectively excited by injecting the optical signal from two ports (marked in Fig. 3a), respectively. In addition, we measure transmission signals from two output ports, which are labeled by the clockwise port and the counterclockwise port, to illustrate the one-way edge propagation in the sample. More details on the experimental measurements can be found in Method. Figure 3c displays the measured transmittance spectra with the CPS being excited. Red and blue lines correspond to averaged transmittance spectra from clockwise and counterclockwise output ports, respectively. Transparent regions around data lines correspond to the fluctuations of measured

transmission signals. It is shown that the measured transmissivity from the clockwise port is much larger than that from the counterclockwise port around the central frequency range in each FSR (highlight by dark regions), showing the excitation of one-way topological edge states. In contrast, as for the case with the input frequency being away from the central frequency, the measured transmissivities from the clockwise output port is significantly decreased and the number of resonant peaks is increased, indicating the excitation trivial bulk/edge eigenmodes. Then, we measure the transmission spectra of the structure by exciting the APS, as shown in Fig. 3d. Contrary to the experimental results of the CPS, the measured transmission from the counterclockwise port is much larger than the clockwise counterpart around the central frequency range in each FSR. These measured transmission spectra for the FHPTI are matched to simulation results in Fig. 2. In addition, due to the large loss effect in the fabricated sample, we find that measured transmissions and topological frequency regions are lower than simulation counterparts. Figure 3e, f presents measured field distributions of topological edge states under the excitations of CPS and APS at 1550.7nm. It is shown that the pseudospin-dependent topological edge transports are observed, as highlighted by white arrows. For comparison, we further measure the field distribution by exciting trivial bulk states at 1551.4nm, as shown in Fig. 3g. A large part of electric fields is penetrated into the bulk, corresponding to the excitation of trivial bulk states. The above phenomena clearly manifest the realization of pseudospin-dependent topological edge states in the FHPTL.

The microscope image of fabricated VHPTI with three layers is shown in Fig. 4a. It is shown that the fabricated sample is consistent with the theoretical design. Similar to the above face-centered condition, we measure transmission spectra by exciting the sample under

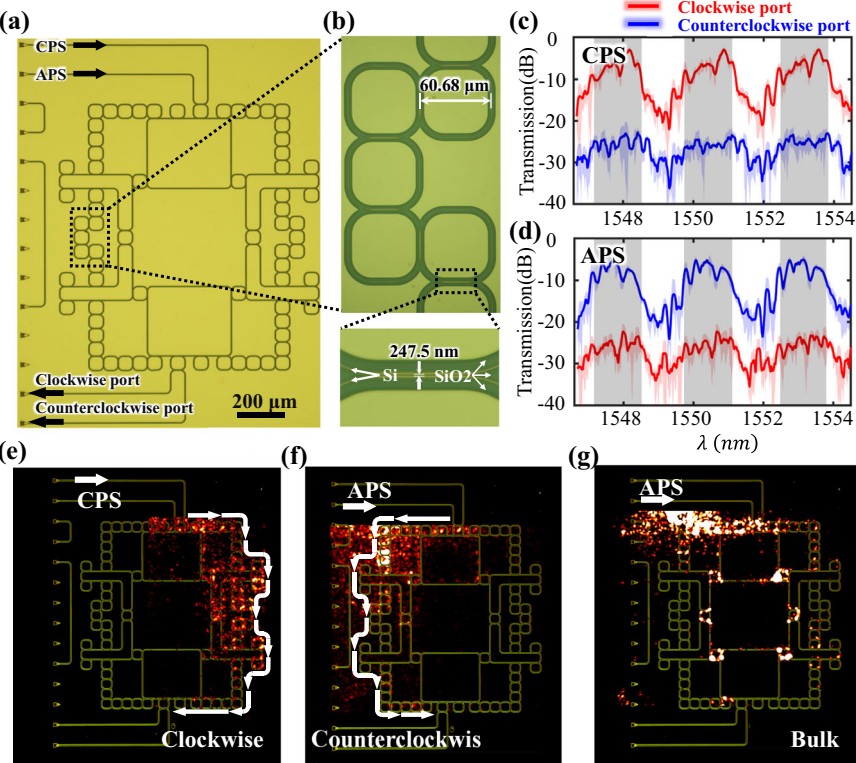

**Fig. 3 | Experimental results of the face-centered hyperbolic photonic topological insulator. a** The microscopy image of the fabricated face-centered hyperbolic sample. **b** The enlarged view of the site optical resonator. **c, d** Measured transmission spectra with the CPS and APS being excited. Red and blue lines correspond to transmission spectra from clockwise and counterclockwise output ports, respectively. **e, f** Measured field distributions of topological edge states with the excited pseudo-spins being CPS and APS in the face-centered hyperbolic photonic topological insulator. **g** Measured field distributions of trivial bulk states in the face-centered hyperbolic photonic topological insulator.

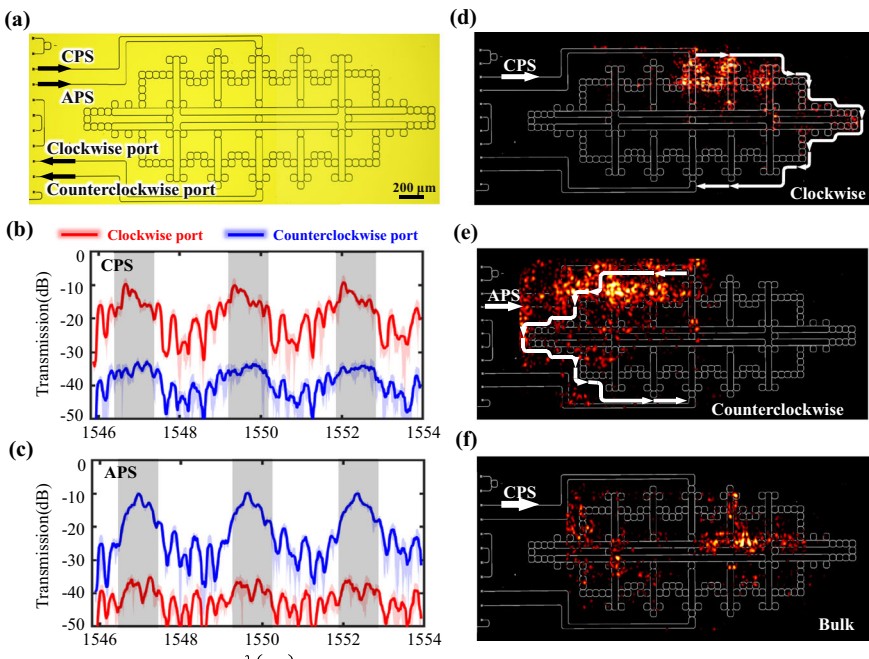

**Fig. 4 | Experimental results of the vortex-centered hyperbolic photonic topological insulator. a** The microscopy image of the fabricated vortex-centered hyperbolic sample. Because the microscope field of view is relatively small, the whole microscope picture is taken by twice independent photographing. And then we joint them together. **b, c** Measured transmission spectra with the CPS and APS being excited. Red and blue lines correspond to transmission spectra related to clockwise and counterclockwise output ports, respectively. **d, e** Measured field distributions of topological edge states with the excited pseudo-spins being CPS and APS in the vortex-centered hyperbolic photonic topological insulator. **f** Measured field distributions of trivial bulk states in the vortex-centered hyperbolic photonic topological insulator.

the CPS and APC, as shown in Fig. 4b, c. Red and blue lines present measured transmissions from clockwise and counterclockwise output ports. It is clearly shown that the pseudospin-dependent one-way edge states around the central frequency range in each FSR also exist in the fabricated VHPTI. Furthermore, we measure the field distributions at 1549.3 nm under the excitations of CPS and APS, as shown in Fig. 4d, e. It is clearly shown that pseudospin-dependent edge propagations appear, as marked by white arrows. The measured field distribution under the excitation of trivial bulk states at 1550.2 nm is plotted in Fig. 4f, where the significant bulk signals appear. These experimental results clearly demonstrate the realization of topological edge states in the three-layer VHPTI.

Finally, we experimentally explore the robustness of topological edge states in the fabricated FHPTI and VHPTI. For this purpose, we introduce the defect into the boundary of FHPTI and VHPTI, as shown in Fig. 5a, b. The defect we studied is caused by the missing of boundary ring resonators, as enclosed by the red dashed box. Such a type of defect has been widely used to demonstrate the robustness of topological edge states in many topological photonic structures[7,30,35,62,63]. We theoretically expect that there is no back-scattering of hyperbolic photonic topological edge states around the defect. This can be confirmed by large-valued transmissivities of topological edge states in both defective and defect-free hyperbolic topological structures, along with analyzing their near-field distributions.

Measured transmission spectra of two-layer FHPTI (three-layer VHPTI) under the excitations of CPS and APS are plotted in Fig. 5c, d, e and f, respectively. It is clearly shown that the pseudospin-selected large transmission still exists in these structures with defects. In particular, under the excitation of CPS (APS) on these two samples, the measured transmissions from the clockwise (counterclockwise) port are much larger than that from the counterclockwise (clockwise) port in the central frequency range of each FSR sustaining

topological edge states (dark regions), being consistent with experimental results without defects. Furthermore, near-field distributions in the FHPTI (VHPTI) under excitations of CPS and APS are also measured at the wavelength of 1550.1 nm (1549.1 nm), as shown in Fig. 5g, h,i, j. We find that the pseudospin-dependent one-way propagations along hyperbolic edges still exist in FHPTI and VHPTI with defects, showing robust boundary propagations in these two fabricated hyperbolic photonic topological insulators. These experimental results are in a good consistence to simulation results (See Supplementary Fig. S9 for details).

## Discussion

In conclusion, we have given the experimental demonstration on the realization of hyperbolic photonic topological insulators by coupled ring-resonator arrays. Both FHPTI and VHPTI have been designed and fabricated on silicon chips. The boundary-dominated optical one-way edge states with pseudospin-dependent propagation directions have been observed in those systems. Furthermore, the robust edge propagations in hyperbolic photonic topological insulators with defects have also been verified. This provides a substantial step toward the investigations of many other topological photonic states in non-Euclidean space, such as the higher-order topological states and non-Hermitian topological states, which are expected to be realized in the future. Besides the conceptual advantage, it is expected that the boundary-dominated topological edge states persist in our designed hyperbolic photonic structure, irrespective of local structural details. Hence, we can achieve increasingly intricate layouts and shapes with boundary-dominated responses. This should have potential applications in the field of designing high-efficient topological photonic devices, such as topological lasers, topological delay lines, topological quantum circuits, and topological quantum sources.

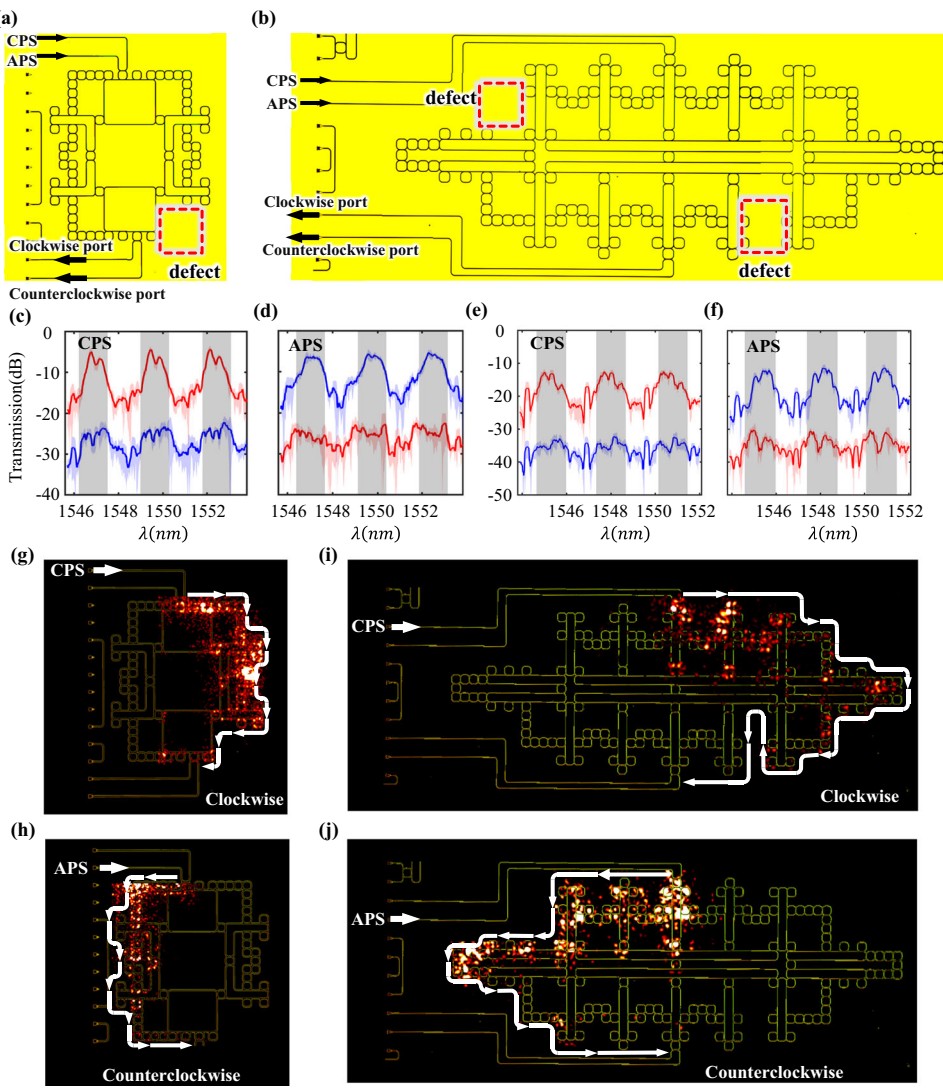

**Fig. 5 | Experimental demonstrations on the robustness of hyperbolic photonic topological insulators. a, b** The microscopy image of the fabricated face-centered and vortex-centered hyperbolic sample with defects. **c, d** The measure transmission spectra of face-centered hyperbolic samples by exciting the CPS and APS, respectively. **e, f** The measure transmission spectra of vertex-centered hyperbolic samples by exciting the CPS and APS, respectively. **g, h** Measured field distributions of topological edge states with the excited pseudo-spins being CPS and APS in face-centered hyperbolic photonic topological insulators. **i, j** Measured field distributions of topological edge states with the excited pseudo-spins being the CPS and APS in the vortex-centered hyperbolic photonic topological insulator.

## Methods

### Sample fabrication

The samples were fabricated using standard complementary metal-oxide-semiconductor processes and 248 nm deep ultraviolet (DUV) lithography processes. The substrate was a silicon-on-insulator wafer with a 220-nm-thick top Si layer. First, a thin oxide layer was formed on the wafer by thermal oxidation. A 150-nm-thick polycrystalline silicon (poly-Si) layer was deposited by low-pressure chemical vapor deposition for the coupling grating fabrication. The wafer was coated with a positive photoresist. The pattern was created by DUV lithography, and then the pattern was transferred onto the poly-Si layer by double inductively coupled plasma etching processes. The etching depth for the sample is 220 nm. A special annealing process was performed to smooth the sidewall of the device. Subsequently, a layer of 1-μm-thick cladding oxide was deposited by plasma-enhanced chemical vapor deposition.

### Experimental measurements

The continuous wave laser (1500–1630 nm) was employed to measure the sample in the experiment. The incident light was first coupled to the single-mode fiber (SMF). And then we used the polarization controllers to adjust the polarization state of the light. Lights entered into the chip by the fiber array. The output signals were collected by another SMF of the fiber array and detected by a high-speed optical power monitor. To sweep the wavelength of the laser, we can obtain the transmission spectrum in the whole near-infrared band. In addition, the light in the silicon waveguides scatters in the vertical direction, thus we can observe the transport of the edge states. We used an optical microscopy system and an infrared InGaAs camera to directly image the edge modes and bulk modes.

It is noted that the experimentally measured losses of the whole system for the 2-layer and 3-layer hyperbolic photonic topological insulators are about -21 dB and -26 dB, respectively. In this case, we can determine the insertion loss of the hyperbolic photonic topological insulators by analyzing the individual insertion losses in each component of the experimental setup, including the coupling between the fiber array and 1D gratings, the on-chip waveguide used for the selectively excitation of APS and CPS modes, and the propagation loss of fibers. To obtain the coupling loss between 1D gratings and the fiber

array, we measure the transmissivity of a test structure, which only consists of an input 1D coupling grating, a short connecting waveguide (130 μm), and an output 1D coupling grating. It is noted that the propagation loss of the short waveguide (2 dB/cm × 130 μm) can be ignored compared to the coupling loss between the fiber array and 1D gratings. Hence, the total coupling loss between input/output gratings and the fiber array is about -14 dB. Moreover, the total length of the silicon waveguide, which is used to selectively excite APS or CPS mode, is about 0.25 cm, and the associated propagation loss is about -0.5 dB (2 dB/cm × 0.25 cm). In addition, the measured propagation loss coming from all fibers is about -1.5 dB. In this case, the total optical loss except for the hyperbolic photonic topological insulators is about -14 dB + 1.5 dB+0.5 dB = 16 dB. After subtracting the loss of -16 dB, the insertion losses for 2-layer and 3-layer hyperbolic photonic topological insulators are about ~5 dB and ~10 dB, respectively.

## Data availability

All data are displayed in the main text and Supplementary Information.

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

## Acknowledgements

This work was supported by the National key R & D Program of China (2022YFA1404904, 2018YFB2200400) (X. Zhang and W. Zhang), Young Elite Scientists Sponsorship Program by CAST (No. 2023QNRC001) (W. X. Zhang), National Natural Science Foundation of China (No.12234004, No.12104041) (X. Zhang and W. X. Zhang), and BIT Research and Innovation Promoting Project (No. 2022YCXY030, No.2023YCXY020) (L. Huang).

## Author contributions

L. Huang and W. X. Zhang finished the theoretical scheme and designed the hyperbolic photonic topological insulator. L. He finished the experiments with the help of H. Zhang, D. Liu, X. Feng, F. Liu, K. Cui, Y. Huang, and W. Zhang. W. X. Zhang, L. Huang, and X. Zhang wrote the manuscript. X. Zhang initiated and designed this research project.

## Competing interests
The authors declare no competing interests.
