## [Peer Review File · Nature Communications]

REVIEWER COMMENTS

Reviewer #1 (Remarks to the Author):

The authors of this manuscript report the experimental realization of hyperbolic topological insulating states in the systems of coupled optical resonators with both face-centered and body-centered structures. Hyperbolic crystallography and, in particular, topological phases in this context, draw some attention in the community. So far, the relevance of this pursuit has been limited to metamaterials. The first experimental observation of a hyperbolic topological insulator has been reported in topolelectric circuits (ref. 47), and so far, this is the only platform where these states have been observed. Now, the present work extends this territory to the photonic metamaterials, which is definitely its main novel aspect. As such, I am inclined to recommend the paper for publication in Nat. Communications after the authors address the following comments.

Being a theorist, I cannot judge the experimental part of the paper. The theoretical part represents a straightforward, albeit cumbersome, application of topological band theory of hyperbolic lattices, which has been recently developed, using the tight-binding formalism. What drew my attention is the analysis of the defect physics which in my opinion deserves more attention. In particular, the following points should be addressed:

1. The authors should clearly state what kind of defects they are studying in their experiment and how are these characterized.
2. What is the theoretical expectation for a particular kind of a defect and how does this compare with the experimental results.

Therefore, I can recommend the paper for publication in Nat. Communications after the authors address these points.

Reviewer #2 (Remarks to the Author):

In this manuscript, the authors design, simulate, and experimentally realize a photonic topological insulator on a so-called hyperbolic lattice using optical ring resonators. Hyperbolic lattices are discrete versions of negatively curved space that recently spurred a lot of attention for their potential

applications in quantum simulation of gravity, quantum error correction, and topological phases of matter. Previously, hyperbolic lattices have been realized experimentally with circuit quantum electrodynamics and topolelectric circuits, and the present work adds another experimental platform for their investigation. In particular, the photonic platform seems to be particularly useful for emulating topological ground states and might have technological applications such as topological lasers.

Let me first summarize what has been accomplished in this work. To realize a finite hyperbolic lattice, both sites/nodes and edges/links are realized with optical rings from waveguides etched onto a silicon surface; some rings acting as sites, others as edges. The coupling between these ring resonators yields an effective tight-binding Hamiltonian that features both nearest-neighbor-site couplings, but also couplings between sites that are further apart. The couplings carry complex phases, which helps in realizing topological states. The authors consider two versions of the hyperbolic lattice, called face-centered and vertex-centered, but I will not address this difference in the following. First the authors numerically simulate the propagation of signals in these geometries. They show the existence of edge-localized modes at certain frequencies that propagate with fixed directionality (depending on their pseudospin), while excitations at other frequencies are pronounced in the bulk. The authors then implement these geometries with an actual experimental and verify their simulation results through measurements. Remarkably, the propagation of signals within the lattice can be visualized in a site-resolved manner. At last, the authors introduce some defects (missing links) in the system and experimentally confirm the robustness of the edge states against this disorder.

I am excited about this work, because it adds a novel experimental platform for realizing topological states of matter in negatively curved space to the toolbox of metamaterials and artificial table-top geometries, and the combination of numerical simulations and actual experiments give a sufficient picture to interpret the experiments. As such, the manuscript bears potential for publication in Nature Communications. However, I believe that some changes to the manuscript are necessary to guarantee the expected high impact of the journal.

1) The manuscript should make clear which tight-binding Hamiltonians can actually be realized with this platform. For instance, Eq. (1) is not quite correct, since, as the authors explain, some hexagons feature NNN and NNNN couplings, and others do not. Hence Eq. (1) that treats all sites equally cannot be correct. Furthermore, how easily can one tune the amplitudes and complex phases of the NN, NNN, NNNN couplings? How would one design the experiment to achieve a certain set of NN, NNN, etc couplings, or is it impossible to change them?

2) Regarding the potential applications of the edge modes: How is this geometry superior to a simple ring geometry?

3) The analysis of the vertex-centered lattice (simulation and experiment) does not add much beyond what is seen in the face-centered one, and vice versa; for instance, the captions of Figs. 3 and 4 are almost identical. Perhaps one could focus on one geometry, and explain the other one in less detail, leaving the details to the supplementary material.

Two more comments:

4) Line 158: In which sense do the counter-propagating modes have opposite Chern numbers? In the figures, all Chern numbers are positive.

5) The editing of the supplementary material is, in my opinion, not up to the standards of the field, with some characters not compiled properly and showing up as boxes (page 4 below Eqs S4 and S5) and generally all special and Greek characters not properly embedded into the text.

Reviewer #3 (Remarks to the Author):

In the present manuscript, the authors experimentally demonstrated on the realization of hyperbolic-lattice photonic topological insulators (PTIs) by coupled optical ring resonators. Both face-centered and vertex-centered hyperbolic PTIs with non-trivial real-space Chern numbers were designed and fabricated in silicon photonics platform. The authors measured transmission spectra and steady-field distributions of hyperbolic PTIs, and thus observed the boundary-dominated one-way edge states with pseudospin-

dependent propagation directions. They also claimed that they have verified the robust edge propagations in hyperbolic PTIs with defects.

The results are solid in both theory and experiment. This work is the first experimental implementation of hyperbolic PTIs as I know, and thus it has certain novelty to develop the emerging field of topological physics and photonics. However, I think the manuscript in this version has not reached the high quality of Nature Communications, due to my comments in the following. I will recommend for reconsideration, if the authors can address my concerns accordingly.

(1) In Lines 39-40, the authors claim topological TIs and semimetals have been experimentally realized by metamaterials and photonic crystals in different dimensions [3-13]. In fact, some of them don't belong to metamaterials and photonic crystals in physics, e.g. refs. 5 and 7. On the other hand, some pioneer works in topological photonic crystals have been missed, except for Hall /spin-Hall phases.

(2) In Lines 75-77, the authors claim 'Our work may have potential applications in designing next-generation topological photonic devices'. What is the meaning of 'next-generation' here?

(3) It is difficult to identify the pink dots in Fig. 1(a).

(4) In experiment, the transmission spectra with defects [Figs. 5(c-f)] is about 5 dB less than the spectra without defects [Figs. 3(c-d) and 4(b-c)]. The authors should seriously discuss the robustness based on their experimental results, since such 5-dB deviation is large in terms of percentage, although the transmission lines have the similarity of the trends.

(5) The efficiency of APS case seems be lower than that of CPS. The authors should give the explanation. Furthermore, how much is the insertion loss of the hyperbolic PTIs in experiment?

Response Letter to Reviewers

We are grateful for the constructive comments on this manuscript (NCOMMS-23-22445-T) from three reviewers.

In the text below, reviewer comments are quoted in **blue** and followed by our detailed response. We have also revised the manuscript and the Supplemental Materials based on the reviewer comments, and these updates are highlighted in **red** in those files. In the text below, these updates are also highlighted in *Italics*.

Response to comments of the reviewer #1

The authors of this manuscript report the experimental realization of hyperbolic topological insulating states in the systems of coupled optical resonators with both face-centered and body-centered structures. Hyperbolic crystallography and, in particular, topological phases in this context, draw some attention in the community. So far, the relevance of this pursuit has been limited to metamaterials. The first experimental observation of a hyperbolic topological insulator has been reported in topoletric circuits (ref. 47), and so far, this is the only platform where these states have been observed. Now, the present work extends this territory to the photonic metamaterials, which is definitely its main novel aspect. As such, I am inclined to recommend the paper for publication in Nat. Communications after the authors address the following comments.

Being a theorist, I cannot judge the experimental part of the paper. The theoretical part represents a straightforward, albeit cumbersome, application of topological band theory of hyperbolic lattices, which has been recently developed, using the tight-binding formalism. What drew my attention is the analysis of the defect physics which in my opinion deserves more attention. In particular, the following points should be addressed:

Therefore, I can recommend the paper for publication in Nat. Communications after the authors address these points.

Reply: We would like to thank the reviewer for the careful review, positive evaluation and valuable suggestions of our work. We have considered and evaluated all reviewer's suggestions in the revised version of our manuscript, particularly regarding the analysis of defect physics. In the following, we respond to each point individually.

1. The authors should clearly state what kind of defects they are studying in their experiment and how are these characterized.

Reply: We would like to thank the reviewer for the comment. The defect we studied is caused by the missing of boundary lattice sites. Such a type of defect has been used to demonstrate the robustness of topological edge state in many previous photonic structures (Refs. [7, 30, 35, 62, 63] in the main text). For a clear illustration, we plot the hyperbolic lattice model with defects, as shown in Fig. R1(a). The missing lattice sites are enclosed by the red dashed box. In experiments, we can characterize such a type of defect in our hyperbolic photonic topological insulator by the number and position for the missed site ring resonators. Figure R1(b) plots the schematic diagram of our hyperbolic photonic structure with defects, where three ring resonators within the red dashed box are missed.

Fig. R1. (a). The defective hyperbolic lattice model, where the light-colored sites enclosed by the red dashed box represent the missing boundary points. (b). The schematic diagram of the corresponding hyperbolic photonic structure, wherein certain boundary resonators are absent.

Action taken:

- ◆ In the revised manuscript, we have added the following discussion in lines 301-305 to clarify boundary defects: “*For this purpose, we introduce the defect into the boundary of FHPTI and VHPTI, as shown in Figs. 5(a) and 5(b). The defect we studied is caused by the missing of boundary ring resonators, as enclosed by the red dashed box. Such a type of defect has been widely used to demonstrate the robustness of topological edge state in many topological photonic structures^{7,30,35,62,63}.*”.

2. What is the theoretical expectation for a particular kind of a defect and how does this compare with the experimental results.

Reply: We would like to thank the reviewer for the comment. We theoretically expect that there is no backscattering of hyperbolic photonic topological edge states around the defect. Such a theoretical expectation can demonstrate the topological robustness of our hyperbolic photonic topological insulator. In experiments, we can confirm our theoretical expectation by comparing the measured transmission spectra and near-field distributions of hyperbolic photonic topological structures with and without defects.

In Figs. R2(a)-(b), we simulated calculate the transmission spectra of hyperbolic photonic topological structures with and without defects. It is found that the large-valued transmissivities in the frequency range of topological edge states (highlighted by the gray domain) still exist in the structure with defects, indicating that there is no backscattering around the defect. The calculated near-field distributions for structures with and without defects are presented in Figs. R2(c)-(d). We can see that the input wave can unidirectionally propagate along the boundary of the structures without backscattering around defects. The experimental measurements of transmission spectra and near-field distributions are shown in Figs. R2(e)-(f) and Figs. R2(g)-(h). These experimental results show that both of hyperbolic structures with and without defects can exhibit backscattering-immune topological edge states, showing the topological robustness of the system.

Fig. R2. (a-b). Simulated transmission spectra without and with defect. (c-d). Experimental transmission spectra without and with defect. (e-f). Simulated results of field distributions by exciting the continuous-wave with the wavelength being 1549.7 nm(the resonant wavelength of site rings) without and with defect. (g-h). Experimental measurements field distributions of topological edge states in the hyperbolic photonic topological insulator without and with defect.

Action taken:

- ◆ In the revised manuscript, we have added the following discussion in lines 305-308 to clarify the theoretical expectation for the influence of defects: “*We theoretically expect that there is no backscattering of hyperbolic photonic topological edge states around the defect. This can be confirmed by large-valued transmissivities of topological edge states in both defective and defect-free hyperbolic topological structures, along with analyzing their near-field distributions.*”

Response to comments of the reviewer #2

In this manuscript, the authors design, simulate, and experimentally realize a photonic topological insulator on a so-called hyperbolic lattice using optical ring resonators. Hyperbolic lattices are discrete versions of negatively curved space that recently spurred a lot of attention for their potential applications in quantum simulation of gravity, quantum error correction, and topological phases of matter. Previously, hyperbolic lattices have been realized experimentally with circuit quantum electrodynamics and topoletric circuits, and the present work adds another experimental platform for their investigation. In particular, the photonic platform seems to be particularly useful for emulating topological ground states and might have technological applications such as topological lasers.

Let me first summarize what has been accomplished in this work. To realize a finite hyperbolic lattice, both sites/nodes and edges/links are realized with optical rings from waveguides etched onto a silicon surface; some rings acting as sites, others as edges. The coupling between these ring resonators yields an effective tight-binding Hamiltonian that features both nearest-neighbor-site couplings, but also couplings between sites that are further apart. The couplings carry complex phases, which helps in realizing topological states. The authors consider two versions of the hyperbolic lattice, called face-centered and vertex-centered, but I will not address this difference in the following. First the authors numerically simulate the propagation of signals in these geometries. They show the existence of edge-localized modes at certain frequencies that propagate with fixed directionality (depending on their pseudospin), while excitations at other frequencies are pronounced in the bulk. The authors then implement these geometries with an actual experiment and verify their simulation results through measurements. Remarkably, the propagation of signals within the lattice can be visualized in a site-resolved manner. At last, the authors introduce some defects (missing links) in the system and experimentally confirm the robustness of the edge states against this disorder.

I am excited about this work, because it adds a novel experimental platform for realizing topological states of matter in negatively curved space to the toolbox of metamaterials and artificial table-top geometries, and the combination of numerical simulations and actual experiments give a sufficient picture to interpret the experiments. As such, the manuscript bears potential for publication in Nature Communications. However, I believe that some changes to the manuscript are necessary to guarantee the expected high impact of the journal.

Reply: We would like to thank the reviewer for the careful review, positive evaluation and valuable suggestions of our work. In the following, we will give a detailed response to all points proposed by the reviewer.

1) The manuscript should make clear which tight-binding Hamiltonians can actually be realized with this platform. For instance, Eq. (1) is not quite correct, since, as the authors explain, some hexagons feature NNN and NNNN couplings, and others do not. Hence Eq. (1) that treats all sites equally cannot be correct. Furthermore, how easily can one tune the amplitudes and complex phases of the NN, NNN, NNNN couplings? How would one design the experiment to achieve a certain set of NN, NNN, etc couplings, or is it impossible to change them?

Reply: We would like to thank the reviewer for the comment. It is true that the original Eq. (1) is not quite correct due to the fact that only a half hexagons feature NNN and NNNN couplings. We modified Eq. (1) as

$$\hat{H} = \sum_i \omega_0 a_i^\dagger a_i + \sum_{\langle i,j \rangle} J e^{i\varphi/3} a_i^\dagger a_j + \sum_{\langle\langle i,j \rangle\rangle_{half}} J e^{i\varphi/6} a_i^\dagger a_j + \sum_{\langle\langle\langle i,j \rangle\rangle\rangle_{half}} J a_i^\dagger a_j + h.c.. \quad (1)$$

where two brackets $\langle\langle \dots \rangle\rangle_{half}$ and $\langle\langle\langle \dots \rangle\rangle\rangle_{half}$ correspond to summations being restricted NNN and NNNN sites within a half number of hexagons for the i -th site.

In addition, hyperbolic photonic structures with different-valued amplitudes and complex phases of NN, NNN, NNNN couplings can be realized by tuning geometric parameters of site and linking rings. We first consider the photonic realization of the hexagon without NNN and NNNN couplings, as shown in Figs. R3(a)-(d). Here, a linking ring can only couple two NN site rings (shown in Fig. R3(a)), where six linking rings are applied to construct the hexagon. The NN coupling amplitude (J) depends on the separation distance between two site rings and the linking ring. A smaller separation distance corresponds to a stronger coupling amplitude. And, the phase accumulation of the wave propagating from one site ring to the other through the linking ring can be used to tune the coupling phase between two site rings. In particular, the phase difference of the wave propagating along top and bottom paths within the link ring ($\Phi_1 - \Phi_2$) is mapped to the effective NN coupling phase (See Supplementary Note 2 for details). Figure R3(b) presents the schematic diagram of the effective NN coupling. In this case, the coupling amplitude can be tuned by changing the separation distance between two site rings and the link ring, as shown in

Fig. R3(c). The coupling phase can be tuned by altering the vertical position of the link ring, as shown in Fig. R3(d).

Fig. R3. A link ring connects two site rings. (a) Schematic diagram of the connection between a ring resonator and the lattice. (b) Schematic diagram of lattice coupling. (c) Schematic diagram illustrating the tune of coupling amplitudes. (d) Schematic diagram illustrating the tune of coupling phase.

Then, we turn to the photonic realization of the hexagon with NNN and NNNN couplings, as shown in Fig. R4. Here, six site rings are coupled by a single linking ring, as depicted in Fig. R4(a). The effective couplings between NN, NNN, NNNN site rings follow similar roles to the case discussed above. Specifically, the coupling amplitude between two site rings are still solely determined by the separation distance between these two site rings and the link ring. The coupling phase is determined by the difference of the phase accumulation with the wave going from a site ring to the other from two paths (clockwise and counterclockwise directions). Therefore, the method of adjusting the NN, NNN, and NNNN coupling amplitudes for six-coupled site rings remains the same to the case with two-site rings. The correspondence coupling phase can be tuned by engineering the propagation phase Φ_i (illustrated in Fig. R4(b)) through the design of relative locations of six site rings and the length of the linking ring (See Supplementary Note 2 for details).

Fig. R4. A link ring connects six site rings. (a) Schematic diagram of the connection between a ring resonator and the lattice. (b) Schematic diagram of lattice coupling.

Action taken:

- ◆ In the revised manuscript, we have added the following discussion in lines 142-146 to illustrate the way to tune the amplitudes and complex phases of NN, NNN, NNNN couplings: *“In this system, we can tune the effective coupling strength J between two site rings by tuning the separation distance between site rings and the link ring. Additionally, the coupling phases between NN, NNN, and NNNN site rings can be adjusted by engineering the phase accumulation of the wave propagating from one site ring to the other through the link ring.”*.
- ◆ In the revised Supplementary Information, we have added the detailed discussion on the method to tune the amplitudes and complex phases of NN, NNN, NNNN couplings: *“Thus, as illustrated in Figs. S4(c) and S4(d), we can tune the coupling strength by adjusting the separation distance between site rings and the link ring, while tuning of coupling phase by altering the vertical position of the link ring.”* in lines 99-101 and *“Therefore, based on the scenario described in Eq. S19, we know that the S -matrix can be tuned by manipulating the separation distance between the site rings and the link rings, thereby controlling the coupling strength J . A smaller separation distance results in a stronger coupling strength J , while a larger separation distance leads to a weaker coupling strength J . Additionally, we can tune the coupling phase $\varphi_{i,j}$ between different site rings by manipulating the propagation phase Φ_i of the light wave within the link rings.”* in lines 150-156.

2) Regarding the potential applications of the edge modes: How is this geometry superior to a simple ring geometry?

Reply: We would like to thank the reviewer for the comment. There are two advantages of topological edge modes compared to a simple ring geometry on some potential applications: (1). Topological edge modes can possess the strong robustness. In this case, we can arbitrarily engineer the shape of the topological cavity and waveguide that facilitate the construction of complex on-chip photonic structures. The design flexibility allows us to better meet specific demands encountered in the chip-scale integration and optimize performance in micro- and nano-optical devices. (2). The separation between two frequency regions supporting topological edge states (the FSR of the small-size site ring) is significantly larger than the separation between two resonance frequencies of a single large ring (FSR of the large ring). This characteristic proves advantageous for various applications that necessitate distinguishable resonances, such as the generation of correlated photon pairs with fixed frequencies in topological quantum sources.

Action taken:

- ◆ In the revised manuscript, we have added the following discussion in lines 340-345 to illustrate the advantage of this geometry: “*Besides the conceptual advantage, it is expected that the boundary-dominated topological edge states persist in our designed hyperbolic photonic structure, irrespective of local structural details. Hence, we can achieve increasingly intricate layouts and shapes with boundary-dominated responses. This should have potential applications in the field of designing high-efficient topological photonic devices, such as topological lasers, topological delay lines, topological quantum circuits, and topological quantum sources.*”.

3) The analysis of the vertex-centered lattice (simulation and experiment) does not add much beyond what is seen in the face-centered one, and vice versa; for instance, the captions of Figs. 3 and 4 are almost identical. Perhaps one could focus on one geometry, and explain the other one in less detail, leaving the details to the supplementary Information.

Reply: We would like to thank the reviewer for the kind suggestion. According to the reviewer's suggestions, in the revised manuscript we have removed the redundant explanations of the vertex-centered grids and leaving them in the Supplementary Note 4.

4) Line 158: In which sense do the counter-propagating modes have opposite Chern numbers? In the figures, all Chern numbers are positive.

Reply: We would like to thank the reviewer for the comment. It is true that the counter-propagating edge modes have opposite Chern numbers. This is because each ring resonator supports two pseudospin components, which circulate in opposite directions within site rings, as shown in Figs. R5(a) and R5(c). Two pseudospins can form the spin-up (clockwise) and spin-down (counterclockwise) topological edge modes, which propagate along opposite directions. It is noted that the effective NN, NNN, and NNN coupling phases in counterclockwise and clockwise subspaces are conjugate to each other. In this case, the effective lattice model of the hexagon feature NNN and NNNN couplings is shown in Fig. R5(b) (Fig. R5(d)) for the clockwise (counterclockwise) mode. It is shown that the arrow directions of complex couplings are opposite for two cases. Therefore, the calculated Chern numbers associated with counterclockwise and clockwise subspaces exhibit opposite signs.

Fig. R5. (a) The schematic diagram of the ring resonator in the spin-up state. (b) The corresponding lattice model and coupling relationship diagram. (c-d) The results of spin-down state.

Action taken:

- ◆ In the revised manuscript, we have added the following discussion in lines 155-164 to illustrate the reason of opposite Chern numbers: *“In addition, it is worth noting that all eigenmodes of a single ring resonator can be categorized into two decoupled subspaces, namely clockwise modes and counterclockwise modes. The coupling strengths between clockwise and counterclockwise modes are extremely weak in our system. Consequently, we can consider clockwise and counterclockwise*

modes as a pair of decoupled pseudospins. In the following, we call these two pseudo-spins as the clockwise pseudo-spin (CPS) and the anticlockwise pseudo-spin (APS). Two pseudo-spins can be suitably excited from two different ports (shown in Fig. 1(f)). Importantly, it should be emphasized that the effective NN, NNN, and NNN coupling phases in counterclockwise and clockwise subspaces are conjugate to each other. Therefore, the calculated Chern numbers associated with counterclockwise and clockwise subspaces exhibit opposite signs.”.

5) The editing of the supplementary material is, in my opinion, not up to the standards of the field, with some characters not compiled properly and showing up as boxes (page 4 below Eqs S4 and S5) and generally all special and Greek characters not properly embedded into the text.

Reply: We would like to thank the reviewer for the comment. We have rechecked and adjusted the formatting issues in the Supplementary Information.

Response to comments of the reviewer #3

In the present manuscript, the authors experimentally demonstrated on the realization of hyperbolic-lattice photonic topological insulators (PTIs) by coupled optical ring resonators. Both face-centered and vertex-centered hyperbolic PTIs with non-trivial real-space Chern numbers were designed and fabricated in silicon photonics platform. The authors measured transmission spectra and steady-field distributions of hyperbolic PTIs, and thus observed the boundary-dominated one-way edge states with pseudospin-dependent propagation directions. They also claimed that they have verified the robust edge propagations in hyperbolic PTIs with defects.

The results are solid in both theory and experiment. This work is the first experimental implementation of hyperbolic PTIs as I know, and thus it has certain novelty to develop the emerging field of topological physics and photonics. However, I think the manuscript in this version has not reached the high quality of Nature Communications, due to my comments in the following. I will recommend for reconsideration, if the authors can address my concerns accordingly.

Reply: We would like to thank the reviewer for the careful review and valuable suggestions, which help us to greatly improve the manuscript. We appreciate the reviewer for agreeing that our manuscript is solid in both theory and experiment, and has certain novelty to develop the emerging field of topological physics and photonics. However, we acknowledge that there were some issues in our original manuscript. Following the comments of the reviewer, we have carefully revised our manuscript. Specifically, we have added new References on pioneer works in topological photonic crystals. We have conducted new measurements to address the discrepancy for experimental results related to samples with and without defects. In addition, we have analyzed the insertion loss for the fabricated hyperbolic photonic topological insulator. In the following, we give detailed responses to each review's comment and hope that our efforts can convince the reviewer for the recommendation of publishing.

(1) In Lines 39-40, the authors claim topological TIs and semimetals have been experimentally realized by metamaterials and photonic crystals in different dimensions [3-13]. In fact, some of them don't belong to metamaterials and photonic crystals in physics, e.g. refs. 5 and 7. On the other hand, some pioneer works in topological photonic crystals have been missed, except for Hall /spin-Hall phases.

Reply: We would like to thank the reviewer for the comment. We agree with the reviewer that not all of the references we cited belong to metamaterials and photonic crystals. We have revised it to: “*For example, topological photonic insulators and semimetals have been experimentally realized.*”. In addition, we have referenced pioneer works related to topological photonic crystals other than Hall/spin-Hall phases. This includes works on valley photonic crystals, higher-order topological states, and others.

Action taken:

◆ In the revised manuscript, we have added follow references: “

14. Ma, T. & Shvets, G. All-Si valley-Hall photonic topological insulator. *New J. Phys.* **18**, 025012 (2016).
15. Dong, J. W., Chen, X. D., Zhu, H., Wang, Y. & Zhang, X. Valley photonic crystals for control of spin and topology. *Nat. materials* **16**, 298-302 (2017).
16. Gao, F., Xue, H., Yang, Z. et al. Topologically protected refraction of robust kink states in valley photonic crystals. *Nat. Phys.* **14**, 140-144(2018).
17. Noh, J., Huang, S., Chen, K. P. & Rechtsman, M. C. Observation of photonic topological valley Hall edge states. *Phys. Rev. Lett.* **120**, 063902(2018).
18. Xie, B. Y., Su, G. X., Wang, H. F. et al. Visualization of higher-order topological insulating phases in two-dimensional dielectric photonic crystals. *Phys. Rev. Lett.* **122**, 233903(2019).
19. Chen, X. D., Deng, W. M., Shi, F. L. et al. Direct observation of corner states in second-order topological photonic crystal slabs. *Phys. Rev. Lett.* **122**, 233902(2019).
20. Cerjan, A., Jürgensen, M., Benalcazar, W. A., Mukherjee, S. & Rechtsman, M. C. Observation of a higher-order topological bound state in the continuum. *Phys. Rev. Lett.* **125**, 213901(2020).
21. Li, M., Zhirihin, D., Gorlach, M. et al. Higher-order topological states in photonic kagome crystals with long-range interactions. *Nat. Photonics* **14**, 89-94(2020).
22. Shalaev, M. I., Walasik, W., Tsukernik, A., Xu, Y. & Litchinitser, N. M. Robust topologically protected transport in photonic crystals at telecommunication wavelengths. *Nat. Nanotechnol.* **14**, 31-34(2019). ”

(2) In Lines 75-77, the authors claim ‘Our work may have potential applications in designing next-generation topological photonic devices’. What is the meaning of ‘next-generation’ here?

Reply: We would like to thank the reviewer for the comment. The term 'next-generation' is used to express our outlook on the high-efficient topological photonic devices with significantly reduced bulk

regions in the future trend. We recognize that this terminology may lead to the misunderstanding. We have provided a more accurate description on the significance of our work.

Action taken:

- ◆ In the revised manuscript, we have changed it to: “*Our work may have potential applications in designing high-efficient topological photonic devices with significantly reduced bulk domains.*” .

(3) It is difficult to identify the pink dots in Fig. 1(a).

Reply: We would like to thank the reviewer for the comment. We have adjusted Fig. 1(a) to make it become clearer, as shown in Fig. R6.

Fig. R6. The illustration of face-centered $\{6, 4\}$ hyperbolic lattice in the Poincaré disk. Sites rings in the first, second, and third layers are represented by red, blue, and pink dots.

(4) In experiment, the transmission spectra with defects [Figs. 5(c-f)] is about 5 dB less than the spectra without defects [Figs. 3(c-d) and 4(b-c)]. The authors should seriously discuss the robustness based on their experimental results, since such 5-dB deviation is large in terms of percentage, although the transmission lines have the similarity of the trends.

Reply: We would like to thank the reviewer for the comment. The large difference between the transmission spectra of photonic samples with and without defects is resulting from the mismatch

between the coupling grating and the fiber during the experimental measurement on the defective sample. Consequently, we recalibrated our experimental setup and conducted new measurements on the transmission spectra of the defective samples. Experimental results presented in Figs. R7(a)-(b) and Figs. R7(c)-(d) correspond the face-centered and vertex-centered samples, respectively. The updated results demonstrate a high level of consistency between the transmission spectra with and without defects.

Fig. R7. The updated figure of Fig. 5(c-f) in the revised manuscript.

Action taken:

- ◆ In the revised manuscript, we have modified the transmission spectra of the defective samples in Fig. 5.

(5) The efficiency of APS case seems be lower than that of CPS. The authors should give the explanation. Furthermore, how much is the insertion loss of the hyperbolic PTIs in experiment?

Reply: We would like to thank the reviewer for the comment. The difference between measured transmissivities with APS and CPS excitations (Figs. 5(c-f) in the main text) is attributed to the inaccuracy of the experimental setup during the testing of defective samples. Specifically, subsequent to measuring defective hyperbolic PTIs under CPS excitation, we reconfigured the experimental setup for APS measurements, resulting in imperfect coupling between the coupling grating and fiber array. This discrepancy led to a lower efficiency in the APS case compared to CPS. In the revised manuscript, we re-measured transmissivities of defective samples and updated our experimental results, as illustrated in Fig. R7. Notably, it was found that transmissions associated with both CPS and APS were nearly equivalent. The slight differences could be attributed to sample asymmetry induced by fabrication errors.

Additionally, the insertion loss of the hyperbolic PTI can be determined by analyzing the individual insertion losses in each component of the experimental setup, including the fibers, 1D gratings, on-chip Si waveguides, and the hyperbolic PTI itself. Firstly, we measure the transmission

spectrum of the entire experimental system which exhibits a loss of approximately ~ 21 dB (2-layer case) and ~ 26 dB (3-layer case). To evaluate the coupling loss between 1D gratings and fiber array, we measure the transmissivity of a test structure consisting only of an input 1D coupling grating, a short connecting waveguide (130 μm), and an output 1D coupling grating, as shown in Fig. R8(a) (the microscope image). The measured transmissivity is plotted in Fig. R8(b). It should be noted that compared to the coupling loss between fiber array and 1D gratings, any propagation loss from this short waveguide (~ 2 dB/cm \times 130 μm) can be neglected. Therefore, the total coupling loss between input/output gratings and fiber array is approximately ~ 14 dB. Moreover, the total length of the silicon waveguide, which is used to selectively excite APS or CPS mode, is about 0.25 cm, and the associated propagation loss is about ~ 0.5 dB (2 dB/cm \times 0.25 cm). Additionally, the measured propagation loss coming from all fibers is about ~ 1.5 dB. In this case, the total optical loss except for the hyperbolic PTIs is about ~ 14 dB + 1.5 dB + 0.5 dB = 16 dB. After subtracting the loss of ~ 16 dB, the insertion losses for 2-layer and 3-layer hyperbolic PTIs are about ~ 5 dB and ~ 10 dB, respectively.

Fig. R8. (a) The microscope image of the structure used to test the coupling loss between 1D gratings and the fiber array. (b). The experimental transmission spectrum of the test structure.

Action taken:

- ◆ In the Methods section of revised manuscript, we have added the following discussion to clarify the insertion loss of fabricated hyperbolic PTIs: *“It is noted that the experimentally measured losses of the whole system for the 2-layer and 3-layer hyperbolic photonic topological insulators are about ~ 21 dB and ~ 26 dB, respectively. In this case, we can determine the*

insertion loss of the hyperbolic photonic topological insulators by analyzing the individual insertion losses in each component of the experimental setup, including the coupling between the fiber array and 1D gratings, the on-chip waveguide used for the selective excitation of APS and CPS modes, and the propagation loss of fibers. To obtain the coupling loss between 1D gratings and the fiber array, we measure the transmissivity of a test structure, which only consists of an input 1D coupling grating, a short connecting waveguide (130 μm), and an output 1D coupling grating. It is noted that the propagation loss of the short waveguide (2 $\text{dB/cm} \times 130 \mu\text{m}$) can be ignored compared to the coupling loss between the fiber array and 1D gratings. Hence, the total coupling loss between input/output gratings and the fiber array is about ~ 14 dB. Moreover, the total length of the silicon waveguide, which is used to selectively excite APS or CPS mode, is about 0.25 cm, and the associated propagation loss is about ~ 0.5 dB (2 $\text{dB/cm} \times 0.25$ cm). Additionally, the measured propagation loss coming from all fibers is about ~ 1.5 dB. In this case, the total optical loss except for the hyperbolic photonic topological insulators is about ~ 14 dB + 1.5 dB + 0.5 dB = 16 dB. After subtracting the loss of ~ 16 dB, the insertion losses for 2-layer and 3-layer hyperbolic photonic topological insulators are about ~ 5 dB and ~ 10 dB, respectively.”.

REVIEWERS' COMMENTS

Reviewer #1 (Remarks to the Author):

I have carefully read the reviewers' comments and the authors' responses. The authors have satisfactorily addressed my questions, which were focused on the defect physics. In particular, they explained clearly the details regarding the implementation and the robustness of the system, due to its nontrivial topology, when the edge missing links were introduced.

Therefore, I can recommend the paper for publication in Nature Communications in its present form.

Reviewer #2 (Remarks to the Author):

The authors have successfully incorporated all my comments in the revised manuscript. This improved both the quality of the analysis and the presentation. I am excited to recommend the revised manuscript for publication in Nature Communications.

Reviewer #3 (Remarks to the Author):

In the revised manuscript, the problems I concern and my suggestions have been addressed accordingly. I will recommend a publication in current version in Nature Communications.

Response to comments of the reviewer #1

I have carefully read the reviewers' comments and the authors' responses. The authors have satisfactorily addressed my questions, which were focused on the defect physics. In particular, they explained clearly the details regarding the implementation and the robustness of the system, due to its nontrivial topology, when the edge missing links were introduced.

Therefore, I can recommend the paper for publication in Nature Communications in its present form.

Reply: We would like to thank the reviewer for recommending our paper for publication in Nature Communications.

Response to comments of reviewer #2

The authors have successfully incorporated all my comments in the revised manuscript. This improved both the quality of the analysis and the presentation. I am excited to recommend the revised manuscript for publication in Nature Communications.

Reply: We would like to thank the reviewer for recommending our paper for publication in Nature Communications.

Response to comments of reviewer #3

In the revised manuscript, the problems I concern and my suggestions have been addressed accordingly. I will recommend a publication in current version in Nature Communications.

Reply: We would like to thank the reviewer for recommending our paper for publication in Nature Communications.